# Medication-related hospital readmissions within 30 days of discharge—A retrospective study of risk factors in older adults

**Maria Glans** [1,2]*, **Annika Kragh Ekstam** [3], **Ulf Jakobsson** [1], **Åsa Bondesson** [1,4], **Patrik Midlöv** [1]

**1** Department of Clinical Sciences Malmö, Center for Primary Health Care Research, Lund University, Malmö, Sweden, **2** Department of Medications, Region Skåne Office for Hospitals in Northeastern Skåne, Kristianstad, Sweden, **3** Department of Orthopaedics, Region Skåne Office for Hospitals in Northeastern Skåne, Kristianstad, Sweden, **4** Department of Medicines Management and Informatics in Skåne County, Malmö, Sweden

* Maria.Glans@med.lu.se

## Abstract

### Background

Previous studies have shown that approximately 20% of hospital readmissions can be medication-related and 70% of these readmissions are possibly preventable. This retrospective medical records study aimed to find risk factors associated with medication-related readmissions to hospital within 30 days of discharge in older adults (≥65 years).

### Methods

30-day readmissions (n = 360) were assessed as being either possibly or unlikely medication-related after which selected variables were used to individually compare the two groups to a comparison group (n = 360). The aim was to find individual risk factors of possibly medication-related readmissions focusing on living arrangements, polypharmacy, potentially inappropriate medication therapy, and changes made to medication regimens at initial discharge.

### Results

A total of 143 of the 360 readmissions (40%) were assessed as being possibly medication-related. Charlson Comorbidity Index (OR 1.15, 95%CI 1.5–1.25), excessive polypharmacy (OR 1.74, 95%CI 1.07–2.81), having adjustments made to medication dosages at initial discharge (OR 1.63, 95%CI 1.03–2.58) and living in your own home, alone, were variables identified as risk factors of such readmissions. Living in your own home, alone, increased the odds of a possibly medication-related readmission 1.69 times compared to living in your own home with someone (p-value 0.025) and 2.22 times compared to living in a nursing home (p-value 0.037).

**Data Availability Statement:** Public access to this data is restricted by the Swedish Authorities (Public Access to Information and Secrecy Act https://www.government.se/information-material/

2009/09/public-access-to-information-and-secrecy-act/). However, data can be made available for research after a special review that includes approval of the research project by both an ethics committee and the authorities' data safety committee. Data access queries are referred to the Swedish Ethical Review Authority (https://etikprovningsmyndigheten.se/) and KVB, the department responsible for patient data disclosures in Region Skåne, Sweden (https://vardgivare.skane.se/kompetens-utveckling/forskning-inom-region-skane/utlamnande-av-patientdata-samradkvb/).

**Funding:** This study was supported by a grant from the ALF funding from Region Skåne (https://www.med.lu.se/intramed/styrning_organisation/ekonomi_alf/alf), awarded to PM. The study was further supported by grants from FoU Kryh and Södra Sjukvårdsregionen (https://sodrasjukvardsregionen.se/doktorandanslag-projektanslag/), awarded to MG. The funders had no role in study design, data collection and analysis, decision to publish, or preparation of the manuscript.

**Competing interests:** The authors have declared that no competing interests exist.

## Conclusion

Possibly medication-related readmissions within 30 days of discharge, in patients 65 years and older, are common. The odds of such readmissions increase in comorbid, highly medicated patients living in their own home, alone, and if having medication dosages adjusted at initial discharge. These results indicate that care planning before discharge and the provision of help with, for example, managing medications after discharge, are factors especially important if aiming to reduce the amount of medication-related readmissions among this population. Further research is needed to confirm this hypothesis.

## Introduction

Readmission to hospital within 30 days of discharge is common [1–4] especially in older adults [2] where hospitalisation also pose a great risk as it could lead to consequences such as infectious disease, confusion, and risk of falling [5]. Previous studies have shown that approximately 20% of hospital readmissions can be medication-related and close to 70% of these readmissions are possibly or definitely preventable [6]. Results are, however, inconsistent regarding risk factors associated with medication-related readmissions [6,7].

Older adults often suffer from several comorbidities thus leading to the use of multiple medications, both known risk factors concerning hospital readmission within 30 days of discharge [8,9]. This increased risk may be due to polypharmacy increasing the risk of medication errors [10] and medication-related problems such as adverse drug reactions [11] or drug-drug interactions [11,12]. The pharmacokinetic and pharmacodynamic changes occurring with older age further increase the risk of such medication-related problems [13] and, with an increase in medications used, follows an increased risk of potentially inappropriate medication therapy [11,14,15]. Since inappropriate medication therapy has been shown to be associated with medication-related admission [16,17], it may also be associated with medication-related readmission.

We have previously shown that the odds of all-cause readmission to hospital, within 30 days of discharge, increase, in older adults ($\geq$ 65 years), if living in the community dependent on home care [9]. This suggests that frailty and lack of supervision could affect the rate of all-cause readmission. In 2016 Olson et al showed that elderly males living alone, with their adult children as caregivers, are at increased risk of readmissions due to high-risk medication regimens [18], which indicates that living arrangements could also affect medication-related readmissions. Older adults living alone have been found to be more prone to medication errors, most likely due to the lack of someone to monitor, assist or remind the older person to take their medications [19]. Further, older adults have been shown to have decreased understanding of medication instructions as well as poor adherence to medication regimens [19]. Considering that poor adherence has been further shown to increase the risk of 30-day readmission in older adults [20], changes made to medications at discharge, as well as living arrangements, could very well affect medication-related readmissions.

Medication-related readmissions to hospital among older adults are common and, in many cases, preventable [21–23]. However, due to differences in methodology in studies investigating such readmissions, the evidence concerning what causes them is inconsistent [6,7]. In this retrospective medical records study, the aim was to find risk factors associated with medication-related readmissions to hospital within 30 days of discharge in older adults ($\geq$65 years).

The study focuses on living arrangements, polypharmacy, potentially inappropriate medication therapy and changes made to medication regimens at discharge.

## Materials and methods

The data in this study were retrieved from a larger study. The method of data collection has been previously described in detail [9].

### Setting

This study was set at the hospital in Kristianstad, which is in the region of Skåne in the south of Sweden. The hospital has 255 beds and provides elective and, predominately, emergency care to the population of Kristianstad and surrounding municipalities.

**Living arrangements.**   While hospital and primary healthcare in Sweden are provided by the regions, the municipalities are responsible for the provision of home care and nursing care. If a patient is in need of intensified municipal care after hospital discharge, a care plan is formed, in collaboration with a municipal nurse, before discharge. The care plan can involve initiation of help with, for example, activities in daily life or help from a nurse with dispensing and administering medications. In some cases, the care plan results in the decision to move the patient to a nursing home, either short-term or permanently.

**Potentially inappropriate medication therapy.**   The Swedish National Board of Health and Welfare have developed indicators of good medication therapy in the elderly [24]. These indicators help prescribers to assess potentially inappropriate medication therapy, including support in choosing medication regimens within certain diagnoses as well as explicit lists of potentially inappropriate medications (PIMs) and fall risk increasing drugs (FRIDs). Further, the indicators recommend avoidance of polypharmacy, especially the use of three or more psychoactive medications.

### Design and study population

The primary study included 720 adults, aged 65 years and older, who were admitted to the hospital in Kristianstad in 2017. Patients were admitted to either of the following departments: Internal medicine, General surgery, Infectious disease, Orthopaedics, Gynaecology, or Ear/Nose/Throat. Following discharge, the study group (n = 360) were readmitted to hospital within 30 days whereas the comparison group (n = 360) was not. Patients could only occur once in the study, either in the study group or in the comparison group, and only the first of multiple unplanned 30-day readmissions was considered. The following patients were excluded: patients where readmission was planned, patients deceased during initial hospital stay, patients readmitted the same day as being discharged and patients going home against medical advice.

In the current study, the patients readmitted within 30 days (n = 360) were assessed as being either possibly or unlikely medication-related readmissions. The assessment was made with the help of the Assessment Tool for identifying Hospital Admissions Related to Medicine (AT-HARM10) [25]. In this assessment tool, a medication-related (re-)admission is defined as a hospital admission where a medication-related problem is either the main admission cause or significantly contributes to admission. Hence, the patient would not have been (re-)admitted without the medication-related problem.

When readmissions had been divided into the groups of possibly and unlikely medication-related readmissions, variables collected in the primary study were used to individually compare the two groups to the comparison group (n = 360). Further, possibly medication-related

readmissions were analysed, within the group, with respect to variables related to living arrangements prior to initial admission.

The primary focus of the study was to find individual risk factors of possibly medication-related readmissions. Since potentially inappropriate medication therapy and polypharmacy are factors that have been previously indicated to play a role in medication-related readmissions, the presence of these factors was of primary interest. Living arrangements and dependency on municipal care prior to initial admission were other factors of primary interest as were changes made to medication regimens at initial discharge.

Potentially inappropriate medication therapy was defined as the use of PIMs, FRIDs and/or the use of three or more psychoactive drugs, as specified by the Swedish National Board of Health and Welfare [24].

## Data collection

The first author (MG) reviewed hospital electronic medical records in a standardised manner to collect the information needed. The review was not blinded but executed as objectively as possible. This procedure has been previously described in further detail [9].

The assessment of the readmissions' relation to medication, according to the assessment tool AT-HARM10 [25], was performed by the first author (MG) in an un-blinded fashion. The assessment tool is comprised of 10 questions, as seen in S1 Appendix. Questions 1–3 are used to identify unlikely medication-related admissions, question 4 states whether it is hinted or stated in the medical record that the admission is medication-related, and questions 5–10 are used to identify possibly medication-related admissions.

The readmissions were reviewed, and the questions were answered, with the help of hospital medical records data. As soon as a "Yes" was retrieved the patient was assigned to either the group of possibly or unlikely medication-related readmissions. If question 4, whether it was hinted or stated in the medical record that the admission was medication-related, was answered "Yes" the assessment was carried on with questions 5–10 until another "Yes" was retrieved, and the identity of the medication-related problem was determined. If all questions were answered "No" the admission was classified as possibly medication-related.

A compilation of potential medication-related readmissions, as suggested by the first author, were hereafter reviewed, revised, and finally approved by an experienced geriatrician (AKE).

As a measure of the patients' comorbidity the Charlson Comorbidity Index [26,27] was used, taking active and previous medical problems into account. The Charlson Comorbidity Index was assessed using information stated in the hospital medical record at the time of initial discharge.

The number of medications used at discharge was calculated in a standardised manner and the variables polypharmacy and excessive polypharmacy were used to describe the regular use of ≥5 or ≥10 medications, respectively. The list of medications was further analysed to determine the use of PIMs, FRIDs or three or more psychoactive drugs as specified by the Swedish National Board of Health and Welfare [24]. To determine the number and type of medication adjustments made during the initial hospital stay, the medications list and documentation in the medical record were assessed.

## Data analysis

Descriptive statistics were calculated for each variable and data for continuous variables are presented as mean and standard deviation (SD) whereas categorical data are reported as proportions (%). A Student's *t*-test was used to compare continuous data between groups while a

$\chi^2$-test or Fishers' exact test was used for comparison of categorical data. To identify variables individually associated with possibly and unlikely medication-related readmissions, as compared to the comparison group, two multiple logistic regression analyses (manual backward) were conducted. Possibly medication-related readmission (0 = No; 1 = Yes) and Unlikely medication-related readmission (0 = No; 1 = Yes) was used as the dependent variable, respectively.

The following variables were used as independent variables in the two multiple regression analyses: Number of admissions in the past 12 months; Help with activities in daily life from municipality; Help with medications from municipality; Length of initial hospital stay five days or longer; Excessive polypharmacy; Number of medications withdrawn; Number of medications started; Number of dosages adjusted; Using three or more psychoactive medications; Using PIMs; Using FRIDs; Day of discharge; Discharging unit; Living arrangements.

The independent variables "Day of discharge", "Discharging unit" and "Living arrangements" (ordinal data) were dummy coded with "Monday-Thursday", "Internal medicine" and "Living in own home, alone" as reference, respectively.

The regression model was controlled for the variables sex, type of admission and Charlson Comorbidity Index (age adjusted for in the Charlson Comorbidity Index). A goodness-of- fit test was carried out on the regression model using the Hosmer and Lemeshow goodness-of-fit test, Cox & Snell $R^2$ and Nagelkerke $R^2$. P-values at $p<0.05$ were considered as significant.

## Ethics statement

The study was approved by the Regional Ethical Review Board in Lund (registration number 2018/326, protocol 2018/4) and administrative permission to access data from medical records was acquired from Region Skåne. Due to lack of contact information, individual informed consent could not be obtained from individual participants. Therefore, prospective participants were informed about the study via an ad in a newspaper covering the geographical area of the study population. The ad explained the aim and approach of the study and included contact information of the first author, encouraging prospective participants to make contact if they did not want to participate or had further questions about the study. Since the study was strictly observational, all presented data are anonymous and no interventions were made that could in any way harm the participants; this procedure was approved by the ethical review board.

## Results

A total of 143 of the 360 readmissions (40%) were assessed as being possibly medication-related, all of which after receiving the answer "Yes" to one of the questions 5–10 of AT-HARM10 (S1 Appendix). In 43% of these readmissions (n = 61), there was also implication in the medical record of this relation. Such notes were most commonly written in internal medical units (83%, p-value 0.003) and patients, who had this relation identified and commented on in their medical records, were most commonly discharged from an internal medicine unit at initial hospital discharge (69%, p-value 0.009).

Patients possibly readmitted due to medications were more often readmitted to internal medicine units than were patients whose readmission was deemed unlikely to be medication-related (71% as compared to 59%, p-value 0.025). Further, patients readmitted due to other causes than medications were more often readmitted to a general surgery unit (15% as compared to 27%, p-value 0.005). There were no significant differences seen between groups in regard to discharging unit at initial hospital discharge.

**Table 1. Patient characteristics prior to initial admission.**

| Characteristic | Comparison group (n = 360) | PMRR (n = 143) | p-value |
|---|---|---|---|
| Age, Mean (SD) | 78 (8) | 80 (8) | **0.028**[a] |
| Age ≥ 75 years, % | 62 | 74 | **0.008**[b] |
| Sex female, % | 52 | 52 | 0.964[b] |
| Number of readmissions 12 months, Mean (SD) | 1.4 (1.0) | 1.9 (1.3) | **<0.001**[a] |
| Charlson Comorbidity Index, Mean (SD) | 6,4 (3,7) | 8,6 (3,9) | **<0.001**[a] |
| **Living arrangements** | | | |
| Living in own home, alone, % | 36 | 53 | **0.001**[b] |
| Living in own home, with spouse/other, % | 53 | 38 | **0.002**[b] |
| Living in nursing home, % | 10 | 9 | 0.743[b] |
| Help with ADL from municipality, % | 12 | 19 | **0.044**[b] |
| Help with medications from municipality, % | 16 | 25 | **0.023**[b] |

Abbreviations: PMRR–Possibly Medication-Related Readmission, SD–Standard Deviation, ADL–Activities in Daily Life.

[a]Student's t-test

[b] $\chi^2$-test.

Significant p-values are indicated in bold.

### Possibly medication-related readmissions

Patients with a possibly medication-related readmission were older, had more comorbidities and were dependent on help from municipality caregivers to a higher degree than patients in the comparison group (Table 1). Further, they used more regular medications and were more often subject to dosage adjustments at initial discharge than were patients in the comparison group (Table 2).

**Classification of medication-related problems.** 141 of the 143 possibly medication-related readmissions were further categorised into categories 5–10 according to AT-HARM10 (two were excluded due to insufficient supporting information). The 141 patients had a total of 196 medication-related problems (1.4 medication-related problem per patient) according to AT-HARM10. Out of the 141 patients, 96 patients (68%) had one medication-related problem while 37 patients (26%) had two, and nine patients (6%) had three medication-related problems. The 196 medication-related problems were distributed, within the categories 5–10 of the AT-HARM10 instrument, as shown in Table 3.

**Variables associated with possibly medication-related readmission.** Our results show that comorbid patients, with multiple admissions in the previous year and an emergency admission at initial hospital stay, are at increased risk of a possibly medication-related readmission (Table 4). Having adjustments made to medication dosages at discharge as well as living in your own home, alone, are other factors that increase the risk of such a readmission (Table 4). Starting new drugs at discharge, on the other hand, decreases the risk of a possibly medication-related readmission within 30 days of discharge, as further seen in Table 4.

Living in your own home, alone, increases the odds of a possibly medication-related readmission 1.69 times as compared to living in your own home with a spouse/other (p-value 0.025). The odds increase 2.22 times when comparing living in your own home, alone, to living in a nursing home (p-value 0.037).

**Living arrangements.** A comparison was made between patients with a possibly medication-related readmission living in their own home, alone (n = 76), prior to initial admission and those living in their own home with a spouse/other (n = 54). Among patients living alone, a majority were women (61% as compared to 41%, p-value 0.026), dependent on help with

**Table 2. Variables related to initial hospital stay and discharge.**

| Variable | Comparison group (n = 360) | PMRR (n = 143) | p-value |
|---|---|---|---|
| Emergency admission, % | 88 | 97 | **0.002**[e] |
| Length of stay 5 days or longer, % | 53 | 63 | **0.039**[e] |
| Discharged from Internal Medicine, % | 56 | 57 | 0.914[e] |
| Discharged from General Surgery, % | 21 | 25 | 0.412[e] |
| Discharged from Infectious Disease, % | 7 | 6 | 0.512[e] |
| Discharged from Orthopaedics, % | 11 | 9 | 0.506[e] |
| Discharged from Gynaecology/Ear Nose Throat, % | 4 | 4 | 0.988[e] |
| **Medications at discharge** | | | |
| Number of medications in total, Mean (SD) | 9.0 (4.9) | 11.5 (5.0) | **<0.001**[d] |
| Number of regular medications, Mean (SD) | 6.5 (3.7) | 8.6 (3.9) | **<0.001**[d] |
| Number of medications as needed, Mean (SD) | 1.8 (1.8) | 2.4 (2.1) | **0.003**[d] |
| Polypharmacy[a], % | 66 | 87 | **<0.001**[d] |
| Excessive polypharmacy[b], % | 20 | 39 | **<0.001**[d] |
| New medications started, % | 79 | 71 | **0.049**[e] |
| Medications withdrawn, % | 23 | 32 | 0.051[e] |
| Dosages adjusted, % | 24 | 42 | **<0.001**[e] |
| Using PIM[c]s, % | 14 | 15 | 0.945[e] |
| Using FRID[c]s, % | 90 | 96 | **0.028**[e] |
| Using 3 or more psychoactive drugs[c], % | 48 | 46 | 0.560[e] |

Abbreviations: PMRR–Possibly Medication-Related Readmission, SD–Standard Deviation, PIM–Potentially Inappropriate Medication, FRID–Fall Risk Increasing Drug.

[a]Defined as the daily use of five medications or more

[b]Defined as the daily use of 10 medications or more

[c]According to the Swedish National Board of Health and Welfare

[d]Student's t-test

[e]$\chi^2$-test Significant p-values are indicated in bold.

**Table 3. Categorisation of medication-related problems in patients possibly readmitted due to medications—As assessed by AT-HARM10.**

| Category and definition[a] | Number of MRPs | Proportion of MRPs (%) | Example |
|---|---|---|---|
| 5. Could side effects of the drugs the patient was taking (prescribed or not prescribed) prior to the hospitalisation have caused the admission? | 97 | 49 | Patient admitted with warfarin- caused bleeding. |
| 6. Are there abnormal laboratory results or vital signs that could be medication-related and could have caused the admission? | 37 | 19 | Patient admitted with hypokalaemia due to diuretics. |
| 7. Was there any medication-medication interaction or medication-disease interaction (i.e. a contraindication) that could have caused the admission? | 8 | 4 | Patient with insufficient antibiotic effect due to interaction between doxycycline and omeprazole. |
| 8. Did the patient have any, previously diagnosed, untreated or undertreated (e.g. too low dose) indications that could have caused the admission? | 46 | 23 | Patient admitted with stroke due to insufficient anticoagulation treatment (PK-INR too low). |
| 9. Was the patient admitted due to a problem with the dosage form or drug formulation? | 1 | 1 | Patient readmitted due to respiratory problems related to incorrect use of inhalers. |
| 10. Is the cause of the admission a response to cessation or withdrawal of drug-therapy? | 7 | 4 | Patient readmitted with oedema due to terminated hydrochloric thiazide treatment. |

Abbreviations: MRP–Medication-Related Problem, i.e.–"id est" meaning "that is", e.g.–"exempli gratia" meaning "for example".

[a]According to the assessment tool AT-HARM10 [25].

**Table 4. Variables associated with possibly medication-related readmission to hospital within 30 days of discharge[a].**

| Variable | OR | 95%CI | p-value |
|---|---|---|---|
| Sex | 0.88 | 0.57–1.36 | 0.568 |
| Emergency admission | 5.13 | 1.70–15.43 | **0.004** |
| Charlson Comorbidity Index | 1.15 | 1.05–1.25 | **0.002** |
| Number of admissions, 12 months | 1.33 | 1.10–1.61 | **0.003** |
| Excessive polypharmacy[b] | 1.74 | 1.07–2.81 | **0.024** |
| New medications started | 0.54 | 0.33–0.88 | **0.014** |
| Dosages adjusted | 1.63 | 1.03–2.58 | **0.038** |
| Living arrangements (Own home, alone) | | | **0.025** |
| Own home, with spouse/other | 0.59 | 0.37–0.94 | **0.025** |
| In nursing home | 0.45 | 0.21–0.95 | **0.037** |

Abbreviations: OR–Odds Ratio, CI–Confidence Interval.

[a]Adjusted for gender, type of admission and Charlson Comorbidity Index (age adjusted within the Charlson Comorbidity Index)

[b]Defined as a regular intake of 10 medications or more.

Hosmer Lemeshow goodness of fit test p-value: 0.565. Cox & Snell $R^2$: 0.142. Nagelkerke $R^2$: 0.204.

Significant p-values are indicated in bold.

medications from municipality (24% as compared to 9%, p-value 0.034). Excessive polypharmacy was more common in the group of patients living alone (43% as compared to 26%, p-value 0.041). The time between admissions was longer (13±9 days as compared to 9±8 days, p-value 0.006) in this group and readmission to the orthopaedics ward was more common (13% as compared to 2%, p-value 0.022) as was readmission due to an unsustainable home situation (26% as compared to 9%, p-value 0.015).

## Unlikely medication-related readmissions

In patients readmitted due to other causes than medications, the main risk factors were an increased Charlson Comorbidity Index (OR 1.17, CI95% 1.08–1.26, p-value <0.001) and the number of admissions to hospital in the past year (OR 1.48, CI95% 1.25–1.76, p-value <0.001). The odds of readmission in this group also increased if the length of initial hospital stay was five days or longer (OR 1.81, CI95% 1.22–2.70, p-value 0.004) and if discharge occurred on a Friday (OR 1.71, CI95% 1.10–2.67, p-value 0.018) or from the general surgery department (OR 1.95, CI95% 1.22–3.13, p-value 0.005).

## Discussion

In this study, 40% of readmissions within 30 days of discharge, in patients aged 65 years and older, were deemed possibly medication-related. The odds of such readmissions increased in comorbid patients living in their own home, alone. Further, using 10 medications or more on a regular basis increased the odds of a possibly medication-related readmission, as did having dosages adjusted at initial hospital discharge. Time between admissions was longer in patients living alone and readmission due to an unsustainable home situation was more common in these patients compared to those living in their own home with someone.

The number of possibly medication-related readmissions in this study was quite high compared to previous Swedish studies where Gillespie et al [22] and Hellstrom et al [28] found a 24% and 13% medication-related readmission rate, respectively. On the other hand, our result

equals that of Bonnet-Zamponi et al [21] and it is even lower than that of Thomas et al [29] where 64% of readmissions were found to be medication-related. However, in all of these studies, solely readmissions to internal medicine units were included whereas our study also included readmissions to other departments (General surgery, Infectious disease, Orthopaedics, Gynaecology, or Ear/Nose/Throat). Further, the above-mentioned studies all used different time frames from discharge to readmission as well as different definitions of medication-related readmissions. None of the studies used the assessment tool AT-HARM10 to define medication-related readmissions. These differences make it hard to compare results between studies looking at medication-related readmissions, as previously pointed out by El Morabet et al [6]. However, our results suggest that a large proportion of readmissions within 30 days, in older adults, are medication-related and that most of these readmissions are to medicine wards.

In this study, living in your own home, alone, was shown to be a major risk factor of possibly medication-related readmissions. While there are previous studies claiming that living alone is a major risk factor of 30-day readmission [30,31], others have not been able to establish such a connection [32]. Neither of these studies, however, look at medication-related readmissions but rather all-cause readmissions after a certain surgical procedure [30], within a certain medical diagnosis [31], or within potentially preventable readmissions according to certain criteria [32].

However, living alone has been previously shown to increase the risk of medication-related health care use after discharge [33] and Olson et al [18] have shown that male older adults living alone, with their adult children as caregivers, are at increased risk of medication-related readmission within 30 days. Their results imply that someone being there at all times is preferable in the case of medication-related readmission [18]. On the same note, Bragstad et al [34], have shown that the odds of managing well after discharge increase if there is someone present in the home at homecoming, as opposed to coming home to an empty house.

Even though we did not further investigate who cared for the patients living alone, the common denominator is the lack of 24-hour supervision and care which, especially after medication dosages being adjusted and in the case of excessive polypharmacy, most probably plays a big part in medication-related readmission within 30 days. Even though these patients, to a large extent, had help from municipality care givers, it is not the same as always having someone there to check on you and make sure you take your medications according to prescription. This lack of supervision may also contribute to patients living alone waiting longer before seeking health care as well as them more often being readmitted due to an unsustainable home situation than those living with someone.

Readmissions, possibly related to medications, were not affected by length of stay or by being discharged on a Friday or from the general surgery department, as were readmissions unrelated to medications and, as seen in our primary study, all-cause readmissions [9]. This suggests that these factors are mainly related to readmissions to other causes than medications, such as infections, aggravation of symptoms or complications after surgery. Medication-related readmissions, on the other hand, seem to be more closely connected to care planning before discharge and the provision of help with, for example, managing medications after discharge. Further research is needed to determine if this is true.

In this study, our main focus was risk factors related to possibly medication-related readmissions within 30 days of discharge, in older adults (≥65 years). Our results imply that a well thought through care plan, formed, and carried out in collaboration with municipality care givers, is important if wanting to avoid such medication-related readmissions. This is true especially in comorbid patients living alone, using 10 medications or more on a regular basis or if dosage adjustments have been made at initial discharge.

## Strengths and limitations

In this retrospective medical records study, we investigated whether living arrangements, polypharmacy, the use of inappropriate medication therapy, or medication adjustments made at discharge affect the occurrence of possibly medication-related readmissions within 30 days of discharge. The variables included in the study were well suited to answer the posed research question and to investigate what distinguished possibly medication-related readmissions from the comparison group, and from readmissions unlikely to be medication-related. No patient occurred more than once in the study, which added to robustness, as did structuring data collection and using standardised instruments such as AT-HARM10 and Charlson Comorbidity Index. The inclusion of patients from different departments (Internal medicine, General surgery, Infectious disease, Orthopaedics, Gynaecology, and Ear/Nose/Throat) added to diversity and generalisability.

However, since this is an analysis of group data from a primary study, a calculation of group size could not be performed beforehand, thus risking inadequate power. Further, even though care was taken in choosing the included variables, we cannot exclude the possibility that differences between groups are due to confounders, collinearity, or interactions. The unblinded review of medical charts and assessment of risk factors could also, even though objectivity was sought at all times, add to a skewed result. Finally, the generalisability of results from the study is limited due to it being conducted in a single hospital.

## Conclusions

According to this study, possibly medication-related readmissions within 30 days of discharge, in patients 65 years and older, are common. The odds of such readmissions are increased in comorbid, highly medicated patients living in their own home, alone. Further, they increase if medication dosages are adjusted at initial hospital discharge.

These results indicate that care planning before discharge and the provision of help with, for example, managing medications after discharge are factors especially important if aiming to reduce the amount of medication-related readmissions in this population. Further research is needed to confirm this hypothesis.

## Supporting information

**S1 Appendix. Assessment tool for identifying Hospital Admissions Related to Medicine (AT-HARM10).** Includes the AT-HARM10 assessment tool, instructions for use and representative examples of when a question should be answered "Yes" or "No".
(DOCX)

## Acknowledgments

We are indebted to Patrick Reilly for his expertise and advice in editing the manuscript as well as to Louise Davstedt for fast and reliable help with attaining medical records.

## Author Contributions

**Conceptualization:** Maria Glans, Annika Kragh Ekstam, Ulf Jakobsson, Åsa Bondesson, Patrik Midlöv.

**Formal analysis:** Maria Glans.

**Funding acquisition:** Maria Glans, Patrik Midlöv.

**Investigation:** Maria Glans.

**Methodology:** Maria Glans, Ulf Jakobsson.

**Resources:** Patrik Midlöv.

**Supervision:** Annika Kragh Ekstam, Ulf Jakobsson, Åsa Bondesson, Patrik Midlöv.

**Validation:** Annika Kragh Ekstam, Ulf Jakobsson, Åsa Bondesson, Patrik Midlöv.

**Writing – original draft:** Maria Glans.

**Writing – review & editing:** Annika Kragh Ekstam, Ulf Jakobsson, Åsa Bondesson, Patrik Midlöv.

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
