## [Decision Letter · Decision Letter 0]

30 Apr 2021

PONE-D-21-08047

Medication-related hospital readmissions within 30 days of discharge - a retrospective study of risk factors in older adults

PLOS ONE

Dear Dr. Glans,

Thank you for submitting your manuscript to PLOS ONE. After careful consideration, we feel that it has merit but does not fully meet PLOS ONE’s publication criteria as it currently stands. Therefore, we invite you to submit a revised version of the manuscript that addresses the points raised during the review process.

We look forward to receiving your revised manuscript.

Kind regards,

Sreeram V. Ramagopalan

Academic Editor

PLOS ONE

Journal Requirements:

2a) If there are ethical or legal restrictions on sharing a de-identified data set, please explain them in detail (e.g., data contain potentially identifying or sensitive patient information) and who has imposed them (e.g., an ethics committee). Please also provide contact information for a data access committee, ethics committee, or other institutional body to which data requests may be sent.

2b) If there are no restrictions, please upload the minimal anonymized data set necessary to replicate your study findings as either Supporting Information files or to a stable, public repository and provide us with the relevant URLs, DOIs, or accession numbers. Please see http://www.bmj.com/content/340/bmj.c181.long for guidelines on how to de-identify and prepare clinical data for publication. For a list of acceptable repositories, please see http://journals.plos.org/plosone/s/data-availability#loc-recommended-repositories.

Additional Editor Comments (if provided):

Reviewers' comments:

Reviewer's Responses to Questions

**Comments to the Author**

1. Is the manuscript technically sound, and do the data support the conclusions?

Reviewer #1: Yes

2. Has the statistical analysis been performed appropriately and rigorously? 

Reviewer #1: Yes

3. Have the authors made all data underlying the findings in their manuscript fully available?

Reviewer #1: No

4. Is the manuscript presented in an intelligible fashion and written in standard English?

Reviewer #1: Yes

5. Review Comments to the Author

Reviewer #1: The authors used electronic health records from a single-center hospital in Sweden to construct a cohort of older (65+ years of age) patients who were hospitalized in 2017 and readmitted within 30 days for a potentially medication-related reason. A comparison cohort was constructed of patients who were also hospitalized in 2017 but were not readmitted within 30 days. The authors assessed independent risk factors for readmissions (either possibly medication-related or unlikely medication-related), with a particular focus on living arrangements and various aspects of medication use. The authors conclude that possibly medication-related readmissions are common and can be reduced by modifying aspects of pre-discharge care and home help.

Overall, the authors provide data that healthcare systems and health policy makers could potentially use in improving care for older patients who are hospitalized and discharged back into the community. While I recommend accepting this manuscript for publication, improvements could be made by addressing the following points.

•It is unclear why the authors did not match the two cohorts by the admitting hospital department (e.g., General Surgery, Gynaecology, etc). It is possible that patients admitted by differing departments have systematically different distribution of risk factors. Inclusion of the distribution of admitting department between the two cohorts in Table 1 would be informative. If differences are stark, then adjustments in the logistic models would be suggested.

•The authors state (Lines 144-145) that data were collected in a standardized way. It would be informative to know how the authors prevented differential assessment of risk factors based on the study outcome (i.e., whether/how the abstractor was blinded to the readmission status)?

•The authors state (Lines 158-159) that if all questions were answered ‘No’ that the admission was classified as possibly medication-related. Should this be ‘unlikely medication related’ instead?

6. PLOS authors have the option to publish the peer review history of their article (what does this mean?). If published, this will include your full peer review and any attached files.

Reviewer #1: **Yes: **Thy Do

---

## [Author Response · Author response to Decision Letter 0]

24 May 2021

Reviewers’ comments

Overall, the authors provide data that healthcare systems and health policy makers could potentially use in improving care for older patients who are hospitalized and discharged back into the community. While I recommend accepting this manuscript for publication, improvements could be made by addressing the following points.

1. It is unclear why the authors did not match the two cohorts by the admitting hospital department (e.g., General Surgery, Gynaecology, etc). It is possible that patients admitted by differing departments have systematically different distribution of risk factors. Inclusion of the distribution of admitting department between the two cohorts in Table 1 would be informative. If differences are stark, then adjustments in the logistic models would be suggested.

Thank you for this insightful comment. Unfortunately, since the study is based on data retrieved from a larger study, we were not able to match the two cohorts. However, since the study is focusing on risk factors of readmission connected to the initial hospital visit, we have compared the groups (readmitted patients vs comparison group and possibly medication-related readmissions vs unlikely medication-related readmissions) regarding discharging unit at initial hospital discharge and found no significant differences between groups. This is now clarified in lines 208-209 as well as in Table 2 in the manuscript.

2. The authors state (Lines 144-145) that data were collected in a standardized way. It would be informative to know how the authors prevented differential assessment of risk factors based on the study outcome (i.e., whether/how the abstractor was blinded to the readmission status)?

Thank you. It would indeed have been good to have blinded the review of medical records as well as the assessment of risk factors, but this was not done in this study. Objectivity was, naturally, sought at all points. We have now tried to clarify this in line 145 and 149. We have also added it as a limitation in the section Strengths and limitations (line 364-366).

3. The authors state (Lines 158-159) that if all questions were answered ‘No’ that the admission was classified as possibly medication-related. Should this be ‘unlikely medication related’ instead?

Thank you. To answer this question we kindly refer to the S1 Appendix – AT-HARM10 where it is stated “If all the questions are answered with “No”, the admission should be classified as P (possibly medication related)”.

However, the instructions to the AT-HARM10 instrument further states: If all the questions are answered "No", the assessment is still indecisive and needs to be examined by an expert panel.

Luckily, in none of the included readmissions all the questions were answered “No”, hence no further examination was needed. We have tried to further clarify this in lines 200-201.

---

## [Editor Report · Decision Letter 1]

27 May 2021

Medication-related hospital readmissions within 30 days of discharge - a retrospective study of risk factors in older adults

PONE-D-21-08047R1

Dear Dr. Glans,

We’re pleased to inform you that your manuscript has been judged scientifically suitable for publication and will be formally accepted for publication once it meets all outstanding technical requirements.

Kind regards,

Sreeram V. Ramagopalan

Academic Editor

PLOS ONE
---

## [Editor Report · Acceptance letter]

31 May 2021

PONE-D-21-08047R1 

Medication-related hospital readmissions within 30 days of discharge – a retrospective study of risk factors in older adults 

Dear Dr. Glans:

I'm pleased to inform you that your manuscript has been deemed suitable for publication in PLOS ONE. Congratulations! Your manuscript is now with our production department. 

Kind regards, 

on behalf of

Dr. Sreeram V. Ramagopalan 

Academic Editor

PLOS ONE